# Continual Pre-Training for Cross-Lingual LLM Adaptation: Enhancing Japanese Language Capabilities

**Kazuki Fujii**[†*] **Taishi Nakamura**[†*] **Mengsay Loem**[†] **Hiroki Iida**[†] **Masanari Ohi**[†]
**Kakeru Hattori**[†] **Hirai Shota**[†] **Sakae Mizuki**[†] **Rio Yokota**[‡] **Naoaki Okazaki**[†]
[†]Department of Computer Science, School of Computing, Tokyo Institute of Technology
[‡]Global Scientific Information and Computing Center, Tokyo Institute of Technology
{kazuki.fujii@rio.gsic, taishi.nakamura@rio.gsic, mengsay.loem@nlp.c
 hiroki.iida@nlp.c, masanari.ohi@nlp.c, kakeru.hattori@nlp.c,
 shota.hirai@nlp.c, sakae.mizuki@nlp.c, rioyokota@rio.gsic,
 okazaki@c}.titech.ac.jp

## Abstract

Cross-lingual continual pre-training of large language models (LLMs) initially trained on English corpus allows us to leverage the vast amount of English language resources and reduce the pre-training cost. In this study, we constructed Swallow, an LLM with enhanced Japanese capability, by extending the vocabulary of Llama 2 to include Japanese characters and conducting continual pre-training on a large Japanese web corpus. Experimental results confirmed that the performance on Japanese tasks drastically improved through continual pre-training, and the performance monotonically increased with the amount of training data up to 100B tokens. Consequently, Swallow achieved superior performance compared to other LLMs that were trained from scratch in English and Japanese. An analysis of the effects of continual pre-training revealed that it was particularly effective for Japanese question answering tasks. Furthermore, to elucidate effective methodologies for cross-lingual continual pre-training from English to Japanese, we investigated the impact of vocabulary expansion and the effectiveness of incorporating parallel corpora. The results showed that the efficiency gained through vocabulary expansion had no negative impact on performance, except for the summarization task, and that the combined use of parallel corpora enhanced translation ability.

## 1 Introduction

Large language models (LLMs) such as ChatGPT have attracted significant attention by demonstrating human-level language understanding and generation capabilities, as well as generalizability to various fields. However, many LLMs, including Llama 2 (Touvron et al., 2023), are primarily trained on English corpora, and their performance in other languages, especially in those with syntactic structures and writing systems greatly different from English, is decreased (OpenAI et al., 2023). Motivated by this performance gap, establishing methods to build LLMs that excel in Japanese (hereafter referred to as Japanese LLMs) is an important research issue in Japan. In particular, since English language resources are outstanding in terms of quality and quantity, insights on effectively utilizing both Japanese and English language resources are in high demand. For example, it is estimated that there are approximately nine times more English web pages than Japanese web pages[1]. However, pre-training from scratch using both Japanese and English language data requires enormous computational resources, making it difficult to acquire insights in a timely manner.

---

[*]Equal contribution.

[1]Statistics of Common Crawl Monthly Archives:
https://commoncrawl.github.io/cc-crawl-statistics

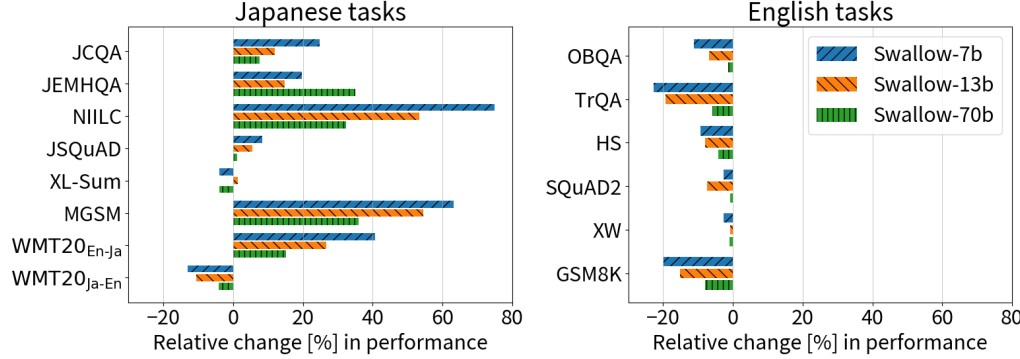

Figure 1: Relative change in performance of `Swallow` compared to `Llama 2`. Japanese tasks (left, see Table 2 for task details) improved by up to approximately 70%.

Therefore, we choose to conduct Japanese continual pre-training from English LLMs, aiming to save computational resources, and obtain insights on how to transfer the knowledge and abilities learned by English LLMs to Japanese. Recently, there has been an increasing number of attempts to adapt LLMs to other languages using continual pre-training (Gupta et al., 2023; Cui et al., 2023; Pires et al., 2023; Zhu et al., 2023; Zhao et al., 2024), but a comprehensive investigation on the effectiveness of continual pre-training has not been conducted. For example, the relationship between the amount of Japanese data used for continual pre-training and model performance, and the impact of the model size on this relationship, are unclear. We construct `Swallow`[2], an LLM with enhanced Japanese capabilities, by performing continual pre-training on Llama 2's 7b, 13b, and 70b models using a Japanese corpus. We then evaluate the performance on six types of tasks including question answering and machine translation in both Japanese and English to analyze changes in knowledge and abilities in both languages. As a result, we demonstrated performance improvements in almost all Japanese tasks for all model sizes (Figure 1).

Furthermore, to identify efficient methodologies for cross-lingual continual pre-training, we also investigate the impact of vocabulary expansion and the effectiveness of Japanese-English parallel corpora. Vocabulary expansion has the effect of shortening token sequence lengths and improving the learning and generation efficiency of Japanese text by adding Japanese characters and words. However, previous studies have not sufficiently analyzed the extra cost of optimizing parameters for the added vocabulary or the impact on performance due to the increase in the amount of learnable text. Therefore, in this study, we conducted a comprehensive evaluation of the impact of vocabulary expansion on the performance of both Japanese and English. Vocabulary expansion improved Japanese text generation efficiency by up to 78%, thanks to a 56.2% reduction in tokens in the Swallow Corpus. This enhancement did not compromise downstream task accuracy, except for summarization.

Parallel corpora are known to have the effect of promoting cross-lingual transfer in multilingual models (Chi et al., 2022; Hu et al., 2020; Feng et al., 2022). However, the effectiveness of parallel corpora in continual pre-training settings of the target language has not been studied in detail. Our experimental results reveal that simply mixing parallel corpora with plain text corpora improves the accuracy of machine translation tasks.

The contributions of this study are as follows:

- `Swallow` achieved the highest performance in Japanese among all models developed in Japan (as of December 2023). We demonstrate that cross-lingual continual pre-training can achieve higher performance with fewer computational resources compared to Japanese LLMs trained from scratch.

---

[2]The LLMs were released with a different official name, but we use this code name for the review.

- We show that continual pre-training is effective for improving Japanese abilities, especially question answering tasks that require Japanese knowledge.

- We provide evidence that the Japanese performance of language models improves monotonically as the amount of Japanese training data increases.

- We show that vocabulary expansion does not affect performance in most tasks, and only degrades performance in automatic summarization.

- We reveal that using parallel corpora together enhances translation ability without affecting the performance of other tasks.

## 2 Related work

### 2.1 Continual pre-training

Continual pre-training is a form of domain adaptation where additional pre-training tasks are performed on a pre-trained language model using text from the target downstream task before fine-tuning the model on that task (Lee et al., 2019; Beltagy et al., 2019; Sung et al., 2019). Since the emergence of open and high-performance English LLMs, there have been an increasing number of attempts to perform continual pre-training to adapt LLMs to other tasks and languages (Gupta et al., 2023; Cui et al., 2023; Pires et al., 2023; Zhu et al., 2023; Zhao et al., 2024). However, a comprehensive investigation of the effects of continual pre-training for various model sizes and training data sizes has not been conducted.

### 2.2 Vocabulary expansion

Vocabulary expansion is a method to increase the vocabulary of a trained LLM. The Japanese writing system differs from English, where kanji characters tend to be converted into UTF-8 byte sequences. For example, a Japanese single-character noun 猫 (cat) is represented by three byte-level tokens <0xE7> <0x8C> <0xAB>, which do not provide any semantic meaning. This treatment not only looks unreasonable as Japanese representation but also increases the sequence length of Japanese text as well as the cost of text generation (Ahia et al., 2023). Adding Japanese characters and words to the vocabulary can reduce the number of tokens required to represent Japanese text and alleviate this problem. In domain adaptation of text embedding models, it is known that adding vocabulary from the target domain can improve performance (Sachidananda et al., 2021; Yao et al., 2021). In contrast, the main motivation for vocabulary expansion in continual pre-training of LLMs is to improve training and generation efficiency in the target language, and there is little knowledge about its impact on performance.

Examples of vocabulary expansion in building Japanese LLMs [3] are limited to evaluating Japanese language ability, and do not evaluate the English language ability. Examples of vocabulary expansion in Chinese LLMs (Cui et al., 2023) only report the performance with vocabulary expansion, and do not compare with a base model. In multilingual LLMs (Ahuja et al., 2023), it was shown that the vocabulary size for each language correlates with performance, but these studies train longer as the vocabulary is expanded, so the isolated impact of vocabulary size remains unclear.

### 2.3 Parallel corpus

In multilingual text embedding models, it has been reported that pre-training with the language modeling objective using parallel corpora mitigates language-specific hidden state vectors and promotes cross-lingual transfer (Chi et al., 2022; Hu et al., 2020; Feng et al., 2022). In applications to LLMs, it has been reported that using parallel sentences for instruction tuning can improve translation ability more efficiently than using the same amount of multilingual corpora (Zhu et al., 2023; Ranaldi et al., 2023). However, the effectiveness of combining continual pre-training with parallel corpora remains unclear.

---

[3] Japanese Stable LM Beta: `https://ja.stability.ai/blog/japanese-stable-lm-beta`

| Params | $d_{model}$ | # Heads | # Layers | Context length | GQA | # Tokens | LR |
|--------|-------------|---------|----------|----------------|-----|----------|-----|
| 7B | 4096 | 32 | 32 | 4096 | ✗ | 100B | $1.0 \times 10^{-4}$ |
| 13B | 5120 | 40 | 40 | 4096 | ✗ | 100B | $1.0 \times 10^{-4}$ |
| 70B | 8192 | 64 | 80 | 4096 | ✓ | 100B | $5.0 \times 10^{-5}$ |

Table 1: Architecture and hyperparameters of the `Swallow` models

# 3 Experimental settings for continual pre-training

The continual pre-training of Swallow involves expanding the vocabulary of a pre-trained `Llama 2` model and then performing continual pre-training using corpora primarily consisting of Japanese. The models are evaluated on six types of tasks in Japanese and English. The evaluation results are compared with LLMs developed in English-speaking regions and in Japan, to demonstrate the comparative effectiveness of our continual pre-training approach. In this section, we describe the training settings, training corpus, vocabulary expansion method, and evaluation method.

## 3.1 Training details

Table 1 shows the hyperparameters of the `Swallow` models. Preliminary experiments were conducted with reference to Llemma (Azerbayev et al., 2024) and Code Llama (Rozière et al., 2023) to determine the parameters. Since continual pre-training is bound by the model architecture of the base model, `Swallow` adopts the same Transformer decoder as `Llama 2`. The hidden size, number of attention heads, number of layers, and context length are the same as `Llama 2`. To maintain consistency with the pre-training phase, a batch size of 1024 was used for all model sizes in `Swallow`, matching the global batch size of 4M tokens in Llama 2.

The AdamW optimizer (Loshchilov & Hutter, 2019) was employed for training the models, with hyperparameters $\beta_1 = 0.9$, $\beta_2 = 0.95$, $\epsilon = 1.0 \times 10^{-8}$. A cosine learning rate scheduler was used, and the learning rate was set to reach its maximum value at 1,000 warmup steps and finally decays to 1/30 of that value. Additionally, a weight decay of 0.1 and gradient clipping of 1.0 were used. Furthermore, Flash Attention 2 (Dao, 2023) was adopted for improved computational efficiency and memory footprints. Refer to Figure 8 in the appendix for the training loss curve.

## 3.2 Training corpora

When continually pre-training with full parameters, forgetting previously learned knowledge is a concern (Jin et al., 2022). One method to prevent forgetting is the Experience Replay technique (Chaudhry et al., 2019). This method involves reusing a portion of the data previously used for training the language model during continual pre-training (Scialom et al., 2022). Following this approach, our study incorporates a portion of the English corpus in addition to the target Japanese corpus for continual pre-training.

The corpus used for continual pre-training includes the Swallow Corpus, which is explained in Appendix A, Japanese Wikipedia[4], and for English, the RefinedWeb (Penedo et al., 2023) and The Pile (Gao et al., 2020). From these corpora, approximately 100B tokens were sampled for the training data of continual pre-training. The sampling was configured so that 5% of the English text comes from RefinedWeb, another 5% from English arXiv paper texts within The Pile, and the remaining 90% from Japanese texts. The Japanese text comprises about 1.6B tokens from Japanese Wikipedia, with the rest from the Swallow Corpus. The ratio of Japanese to English data in the training set was decided based on preliminary experiments (see Appendix B for details).

---

[4]`https://dumps.wikimedia.org/other/cirrussearch/20230320`: Dump dated March 20, 2023.

### 3.3 Vocabulary expansion

In this study, we aim to adapt LLMs to languages with different writing systems from the Latin alphabet. For this purpose, we performed vocabulary expansion. Vocabulary expansion refers to the post-hoc addition of vocabulary to an existing LLM. This increases the amount of text that can be trained and generated within the same computational budget, thereby improving computational efficiency. In the vocabulary expansion adopted in Swallow, we constructed Japanese vocabulary, initialized vectors, and added string preprocessing.

We conducted preliminary experiments to determine the optimal vocabulary size by comparing sequence lengths with 16k, 32k, and 48k Japanese vocabularies. The sequence lengths were 56.2%, 54.1%, and 53.2%, respectively, compared to using original LLaMA vocabulary. Since there was no significant performance difference in Japanese tasks among these sizes, we selected the 16k vocabulary for its advantage in training speed. The construction of Japanese vocabulary involved creating a vocabulary (up to 16k) using the BPE algorithm on the Swallow Corpus segmented by MeCab[5] and the UniDic dictionary[6]. We then merged the Japanese vocabulary (subwords) with the original LLaMA vocabulary, resulting in a total vocabulary size of 43,176. Refer to Appendix E.1 for further details on the vocabulary expansion procedure.

The vectors for the embedding and output layers of the added subwords were initialized with the average of the vectors of the subwords segmented by the LLaMA tokenizer, i.e., the subwords trained by Llama 2, following previous research (Yao et al., 2021). This initialization method was adopted because we confirmed that random initialization led to corrupted outputs.

String preprocessing was enhanced by adding NFKC normalization to utilize the pre-trained knowledge of alphanumeric characters and symbols in the ASCII code range.

### 3.4 Evaluation method

The evaluation methods for Japanese and English are shown in Tables 2 and 3, respectively. The dataset consists of five types of Japanese and four types of English tasks, with few-shot settings for question answering (QA), reading comprehension (RC), automatic summarization (AS), arithmetic reasoning (AR), commonsense reasoning (CR) and machine translation (MT). For details, see Appendix G. We evaluate these downstream tasks under few-shot In-Context Learning without fine-tuning the model. Additionally, the evaluation datasets were not included in our continual pre-training corpora to ensure an unbiased assessment of the model's performance. For all tasks, higher scores indicate better performance.

The selection of tasks was based on discussions in LLM-jp (Han et al., 2024) and the methodology of the Llama 2 paper (Touvron et al., 2023), with an active adoption of tasks related to inference and text generation. The natural language inference task in llm-jp-eval was excluded from evaluation due to unstable scores in the 7b and 13b models (see Appendix F).

## 4 Results

### 4.1 Effects of continual pre-training

Table 4 shows the evaluation results of Swallow and its base model Llama 2 on Japanese and English tasks. Figure 1 displays the increase or decrease rate of Swallow's scores compared to Llama 2. The average score of Swallow on Japanese tasks surpasses Llama 2 by approximately 7 points. On the other hand, the English scores are 2–5 points lower, but the performance drop tends to be smaller as the model size increases. When looking at indi-

---

[5]https://taku910.github.io/mecab/

[6]https://clrd.ninjal.ac.jp/unidic/

[7]https://github.com/Stability-AI/lm-evaluation-harness

| Benchmark | llm-jp-eval (Han et al., 2024) (v1.0.0) | | | JP LM Evaluation Harness[7] (commit #9b42d41) | | | | |
| --- | --- | --- | --- | --- | --- | --- | --- | --- |
| Eval. task | Question Answering | | | RC | AS | AR | Machine Translation | |
| Dataset | JCQA | JEMHQA | NIILC | JSQuAD | XL-Sum | MGSM | WMT'20$_{En-Ja}$ | WMT'20$_{Ja-En}$ |
| Instances | 1,119 | 120 | 198 | 4,442 | 766 | 250 | 1,000 | 993 |
| Few-shots | 4 | 4 | 4 | 4 | 1 | 4 | 4 | 4 |
| Eval. metric | EM acc. | Char-F1 | Char-F1 | Char-F1 | ROUGE-2 | EM acc. | BLEU | |

Table 2: Japanese datasets. Acc. stands for accuracy, EM for exact match. Evaluation datasets include JCQA for JCommonsenseQA (Kurihara et al., 2022), JEMHQA for JEMHopQA (Ishii et al., 2023), NIILC (Sekine, 2003), JSQuAD (Kurihara et al., 2022), XL-Sum (Hasan et al., 2021), MGSM (Shi et al., 2023), and WMT'20 (Barrault et al., 2020).

| Benchmark | LM Evaluation Harness (Gao et al., 2022) (v0.3.0) | | | | | |
| --- | --- | --- | --- | --- | --- | --- |
| Eval. task | QA | | RC | CR | | AR |
| Dataset | OBQA | TrQA | SQuAD2 | HS | XW | GSM8K |
| Instances | 500 | 17,944 | 11,873 | 10,042 | 2,325 | 1,319 |
| Few-shots | 8 | 8 | 8 | 8 | 8 | 8 |
| Eval. metric | acc. | EM acc. | EM acc. | acc. | acc. | EM acc. |

Table 3: English datasets. OBQA for OpenBookQA (Mihaylov et al., 2018), TrQA for TriviaQA (Joshi et al., 2017), HS for HellaSwag (Zellers et al., 2019), XW for XWINO (Tikhonov & Ryabinin, 2021), SQuAD2 (Rajpurkar et al., 2018), and GSM8K (Cobbe et al., 2021).

vidual tasks[8], Japanese question answering (JCQA, JEMHQA, NIILC) shows a significant improvement of up to 75%, and arithmetic reasoning (MGSM) improves by 36–63%. In contrast, English question answering (TrQA) and arithmetic reasoning (GSM8K) degrade by 6–23%. The change in automatic summarization (XL-Sum) is less than 5%. Machine translation shows contrasting results depending on the direction, with a 15–41% improvement in English-to-Japanese (En-Ja) and a 4–13% degradation in Japanese-to-English (Ja-En). Japanese reading comprehension (JSQuAD) has limited room for improvement as `Llama 2`'s score is above 0.8, resulting in less than 10% improvement.

We analyze the impact of continual pre-training on Japanese abilities and knowledge. In the case of arithmetic reasoning (Ja: MGSM, En: GSM8K), while `Llama 2` shows superiority in English (GSM8K > MGSM), `Swallow` improves MGSM, but GSM8K degrades to a similar level, suggesting that the reasoning ability in English has not been fully transferred to Japanese. Considering the reports of reasoning ability transfer in instruction tuning (Ye et al., 2023), combining instruction datasets could be a promising strategy to bring Japanese arithmetic reasoning to the same level observed in English.

Regarding knowledge, the significant improvement in question answering suggests that the acquisition of Japanese knowledge has progressed. Figure 2 illustrates the impact of continual pre-training on the scoring of each question in the NIILC QA dataset. The dense color in the upper left corner indicates that many questions have shifted from incorrect to correct answers, while the opposite is uncommon. This trend suggests that continual pre-training successfully incorporated new knowledge and corrected inaccurate responses.

## 4.2 Comparison with full-scratch models

Table 5 shows the evaluation results of the `Swallow` and major Japanese LLMs trained from scratch (see Table 12 in the appendix for model references). Note that these models are all general-purpose language models without instruction tuning. Compared to the LLMs trained from scratch in Japan (calm-7b, llm-jp-13b-v1.0, PLaMo-13b) (see Appendix I.1 for details of each model), the average score of `Swallow` is 8.4 to 17.4 points higher, demonstrating the usefulness of continual pre-training.

---

[8]We assess performance changes on a relative scale, considering the differences in task score levels.

| | Evaluation in Japanese | | | | | | | |
|---|---|---|---|---|---|---|---|---|
| Model | JCQA | JEMHQA | NIILC | JSQuAD | XL-Sum | MGSM | En-Ja | Ja-En | Avg |
| Llama 2-7b | 38.5 | 42.4 | 34.1 | 79.2 | 19.1 | 7.6 | 17.8 | 17.4 | 32.0 |
| Swallow-7b | 48.1 | 50.8 | 59.7 | 85.7 | 18.3 | 12.4 | 25.1 | 15.1 | 39.4 |
| Llama 2-13b | 70.0 | 44.2 | 41.7 | 85.3 | 21.4 | 13.2 | 21.5 | 19.8 | 39.6 |
| Swallow-13b | 78.4 | 50.6 | 64.0 | 90.1 | 21.7 | 20.4 | 27.2 | 17.7 | 46.3 |
| Llama 2-70b | 86.9 | 46.6 | 52.6 | 90.8 | 23.6 | 35.6 | 26.4 | 24.0 | 48.3 |
| Swallow-70b | 93.5 | 62.9 | 69.6 | 91.8 | 22.7 | 48.4 | 30.4 | 23.0 | 55.3 |

| | Evaluation in English | | | | | | |
|---|---|---|---|---|---|---|---|
| Model | OBQA | TrQA | HS | SQuAD2 | XW | GSM8K | Avg |
| Llama 2-7b | 35.8 | 62.7 | 58.6 | 32.1 | 90.5 | 14.1 | 49.0 |
| Swallow-7b | 31.8 | 48.4 | 53.1 | 31.3 | 88.2 | 11.3 | 44.0 |
| Llama 2-13b | 37.6 | 72.6 | 61.5 | 36.8 | 91.4 | 24.0 | 54.0 |
| Swallow-13b | 35.0 | 58.5 | 56.6 | 34.1 | 90.8 | 20.4 | 49.2 |
| Llama 2-70b | 42.8 | 82.4 | 67.4 | 37.7 | 92.9 | 52.8 | 62.7 |
| Swallow-70b | 42.2 | 77.6 | 64.6 | 37.5 | 92.0 | 48.7 | 60.4 |

Table 4: Evaluation of `Swallow` and its pre-training source, `Llama 2`, in Japanese and English.

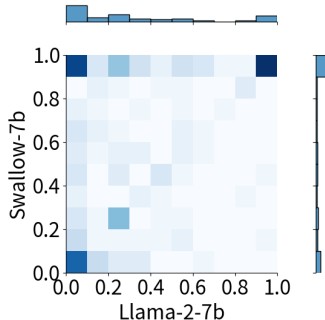

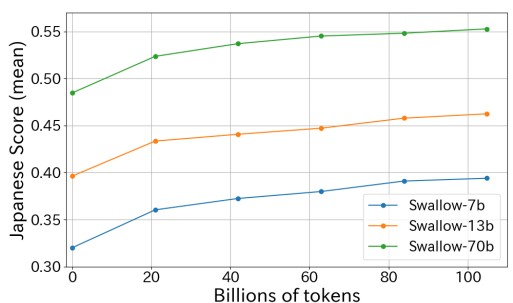

Figure 2: Joint distribution of `Llama 2` (x-axis) and `Swallow` (y-axis) scores (character F1, with 1.0 representing an exact match) for NIILC questions.

Figure 3: Scalability of continual pre-training on Japanese tasks. Score at 0B tokens corresponds to the baseline performance of the `Llama 2` model.

### 4.3 Scalability to training tokens

Figure 3 depicts the relationship between the volume of training data (number of tokens) used for continual pre-training and the average performance score on the Japanese benchmark. We evaluated the performance of `Swallow` 7b, 13b, and 70b at each interval of approximately 20B tokens of the training data. The figure reveals a monotonic upward trend in average scores in correlation with the augmentation of Japanese training data for continual pre-training. Notably, the largest performance increase occurs in the initial training stage with 20B tokens, with subsequent gains diminishing. However, performance consistently improves as the amount of training data increases, suggesting that the performance improvement has not saturated even with approximately 100B tokens of continual pre-training.

| Model | Evaluation in Japanese | | | | | | | | |
| | JCQA | JEMHQA | NIILC | JSQuAD | XL-Sum | MGSM | En-Ja | Ja-En | Avg |
|---|---|---|---|---|---|---|---|---|---|
| calm2-7b | 22.0 | 50.5 | 50.7 | 78.0 | 2.3 | 6.0 | 23.5 | 15.0 | 31.0 |
| Swallow -7b | **48.1** | **50.8** | **59.7** | **85.7** | **18.3** | **12.4** | **25.1** | **15.1** | **39.4** |
| llm-jp-13b-v1.0 | 22.6 | 47.9 | 38.6 | 77.4 | 10.8 | 2.4 | 19.6 | 11.9 | 28.9 |
| PLaMo-13b | 22.7 | **51.9** | 41.4 | 76.2 | 10.3 | 3.6 | 15.8 | 12.0 | 29.2 |
| Swallow -13b | **78.4** | 50.6 | **64.0** | **90.1** | **21.7** | **20.4** | **27.2** | **17.7** | **46.3** |

Table 5: Comparison of evaluation results between `Swallow` and models trained from scratch on Japanese tasks.

## 5 Analysis

### 5.1 Vocabulary expansion

#### 5.1.1 Experimental setup and motivation for vocabulary expansion

To investigate the impact of vocabulary expansion (VE) — specifically, the addition of Japanese words and characters — we compare models pre-trained with and without VE. The model pre-trained without VE is denoted as ¬VE hereafter. Incorporating VE can shorten token sequences, thereby enhancing the efficiency of both training and generating Japanese text. However, the impact of vocabulary expansion on performance is unclear. Intuitively, if the added vocabulary is poorly optimized, it may degrade performance. Yet, studies on domain adaptation scenarios report that VE enhances performance (Sachidananda et al., 2021; Yao et al., 2021). Additionally, being able to train with more text within the same computational budgets could positively influence performance.

#### 5.1.2 Impact of vocabulary expansion

Figure 4 shows the performance of `Swallow` relative to `Swallow`¬VE, where the vocabulary expansion was not applied to the latter. Regarding Japanese language capabilities, the overall impact of vocabulary expansion on performance is minor. Appendix H provides the absolute scores and a more detailed analysis. When examining specific tasks, we observe the performance change in question answering approximately ±10%, yet there is no consistent trend of either improvement or degradation across different model sizes. Therefore, the increase in the amount of training text brought from the vocabulary expansion does not directly affect performance. Additionally, from Figure 7, we did not observe any significant difference in convergence characteristics in the performance curve of Japanese question-answering task. In contrast, automatic summarization (XL-Sum) degraded by about 5–15% with vocabulary expansion for all model sizes[9]. This suggests that the impact of vocabulary expansion may be more apparent in tasks that involve processing longer contexts.

### 5.2 Parallel corpus

#### 5.2.1 Experimental setup for parallel corpus

In the experiments to investigate the effectiveness of parallel corpora, we use `Swallow`¬VE as the baseline to separate the impact of vocabulary expansion, and examine the performance when incorporating parallel corpora in the continual pre-training. We used JParaCrawl 3.0 (Morishita et al., 2022) corpus, which contains approximately 22 million Japanese-English parallel sentences extracted from the web.

We investigate three different settings to utilize a parallel corpus, as shown in Table 6. These methods vary by training sequence and task format. The training sequence can be "two-

---

[9]This trend remained unchanged after instruction tuning, so it does not seem to be a problem with instruction following ability.

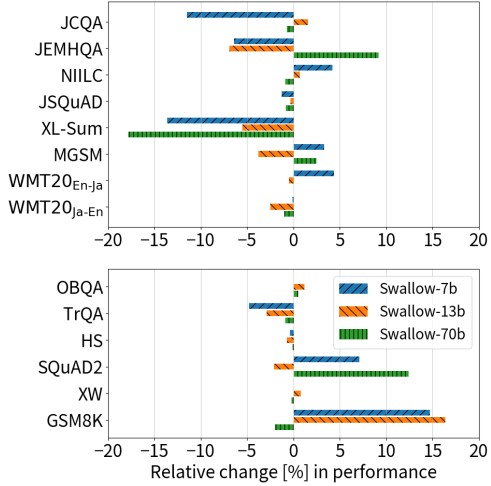

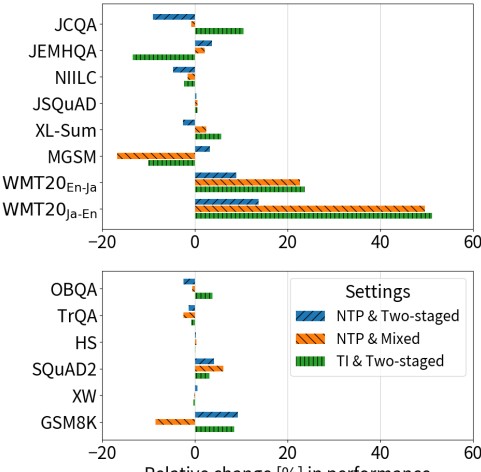

Figure 4: Relative change in performance with versus without vocabulary expansion (Swallow vs. Swallow¬VE).

Figure 5: Relative change in performance when using parallel corpus compared to Swallow¬VE.

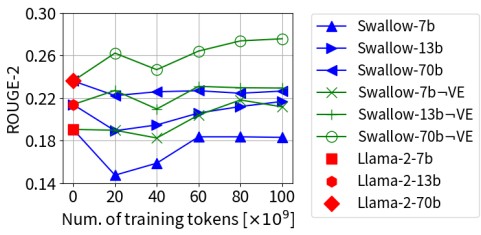

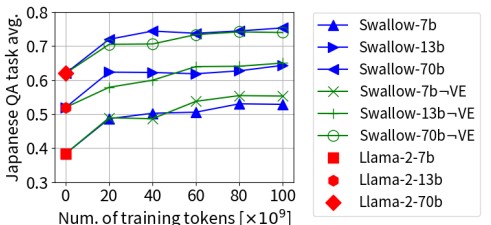

Figure 6: Performance curve of Swallow¬VE and Swallow in automatic summarization (XL-Sum).

Figure 7: Performance curve of Swallow¬VE and Swallow in the average score of question answering task (JCQA, JEMHQA, NIILC).

staged", where the parallel corpus is used first and followed by the multilingual corpus, or "mixed", where it is combined with the multilingual corpora from the start. The intention behind the two-staged setting is the assumption that parallel sentences could facilitate the switch from English-centric to Japanese-centric training.

For task formats, we have "next token prediction" (NTP), which involves concatenating parallel sentences, and "translation instruction" (TI), where the target sentence is predicted based on the source sentence and a translation instruction. Both formats are applied in two directions, Ja→En and En→Ja, for each pair of parallel sentences. The following templates show how parallel sentences are converted into NTP format (top) and TI format (bottom). Note that for the TI format, only the target language sentence is the target for training.

```
[Japanese sentence] [English sentence]
[English sentence] [Japanese sentence]
```

```
Please translate the following Japanese text into English.
[Japanese sentence] [English sentence]
Please translate the following English text into Japanese.
[English sentence] [Japanese sentence]
```

| Task format | Training seq. | Number of training tokens [$\times 10^9$] |
|---|---|---|
| Next Token Prediction (NTP) | Two-staged | 5.6 |
| Next Token Prediction (NTP) | Mixed | 5.6 |
| Translation Instruction (TI) | Two-staged | 2.8 |

Table 6: List of experimental settings in the use of the parallel corpus.

### 5.2.2 Effectiveness of parallel corpus

Figure 5 shows the rate of increase or decrease in scores when using a parallel corpus in conjunction with continual pre-training, with Swallow¬VE without vocabulary expansion as the baseline. Translation performance improved by 9–24% for En-Ja and 14–51% for Ja-En. The improvement in Ja-En is particular to the parallel corpus. Regarding the usage of the corpus, we found that the next token prediction format with a "mixed" setting or the translation instruction format in a "two-staged" setting was effective. In other words, we proved that simply mixing parallel sentences into the multilingual corpus and conducting continual pre-training can effectively improve translation performance. This finding is consistent with the claim that the translation capabilities of LLMs stem from parallel sentences incidentally present in plain text corpora (Briakou et al., 2023).

The relative change in scores for tasks other than translation was within ±15%, without showing consistent improvement or decline across different model sizes. Therefore, no evidence was obtained that the parallel corpus promotes cross-lingual transfer and improves abilities other than translation.

## 6 Conclusions

In this study, we developed Swallow, leveraging Llama 2 models for enhanced Japanese language performance through continual pre-training on Japanese datasets, and analyzed their performance to address the lack of comprehensiveness in model size, training data size, and evaluation methods in previous studies. Through our evaluation, we found that continual pre-training significantly boosts Japanese abilities, particularly in knowledge-intensive question answering task. Consequently, we demonstrated that continual pre-training is an efficient approach for achieving high performance, as the continual pre-trained models outperform Japanese LLMs that are trained from scratch. We also observed that performance improves in line with increases in training data. Furthermore, in our quest for a more efficient methodology, we investigated how expanding the vocabulary and incorporating parallel corpora into continual pre-training affect performance. We revealed that while the vocabulary expansion improves computational efficiency, it has little impact on performance except for summarization. Additionally, we found that simply integrating parallel sentences into plain text corpora improves translation performance, particularly for Japanese-English, but it does not hurt the performance of other tasks. The experimental results and analyses in this study provide important insights into the effectiveness of continual pre-training, the impact of vocabulary expansion, and the effects of using parallel corpora in the development of LLMs for non-English languages.

## 7 Ethical considerations

`Swallow` is subject to the same well-recognized limitations of other LLMs, including the inability to update information after the pretraining phase, the potential for non-factual generation such as unqualified advice, and a propensity towards hallucinations.

Similar to other large language models, `Swallow` may produce content that is harmful, offensive, or biased due to being trained on datasets derived from publicly available online sources. We commit to ongoing fine-tuning and plan to release updated versions of `Swallow` as we make further advancements in resolving these issues.

## 8 Reproducibility statement

All models developed in this study (continual pre-training on Llama 2 7B, 13B, 70B) have already been released on Hugging Face. The benchmark datasets used in this study are also publicly available. Therefore, it is straightforward to reproduce our experimental results reported in Table 4, 5, 11.

## Acknowledgments

This work was supported by the *ABCI Large-scale Language Model Building Support Program* of the AI Bridging Cloud Infrastructure (ABCI), built and operated by the National Institute of Advanced Industrial Science and Technology (AIST). We would like to thank Mr. Takuya Akiba of *Sakana AI* for providing valuable advice on training. We also received advice on vocabulary expansion methods from Mr. Tatsuya Hiraoka of *MBZUAI*.

In the evaluation experiments of the trained LLMs, we utilized data and insights developed and made publicly available by *LLM-jp*. This research was supported by JST, JPMJCR2112.

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

## A   Swallow corpus

Swallow Corpus was constructed by extracting and refining Japanese text from the Common Crawl archives, which consist of 21 snapshots collected between 2020 and 2023, comprising approximately 63.4 billion pages. The resulting corpus contains about 312.1 billion characters (approximately 173 million pages).

## B    Ratio of training data

The ratio of mixing Japanese and English texts was determined through preliminary experiments. We tested two experimental settings with a JA:EN ratio of 5:5 and 9:1, and compared the average performance on Japanese tasks after training on approximately 20B tokens. In the preliminary experiments, the 9:1 setting slightly outperformed the 5:5 setting in terms of the average score on Japanese tasks, hence we adopted the 9:1 JA:EN ratio. While the goal was to construct a powerful LLM specialized for the target language, transferring knowledge from English was also one of the objectives. It has been shown that including 1% of the pre-trained data in training can prevent catastrophic forgetting (Scialom et al., 2022), but due to the unavailability of Llama 2's training data and budgetary constraints, the preliminary experiments were conducted only with the 5:5 and 9:1 ratios.

## C    Training loss curves

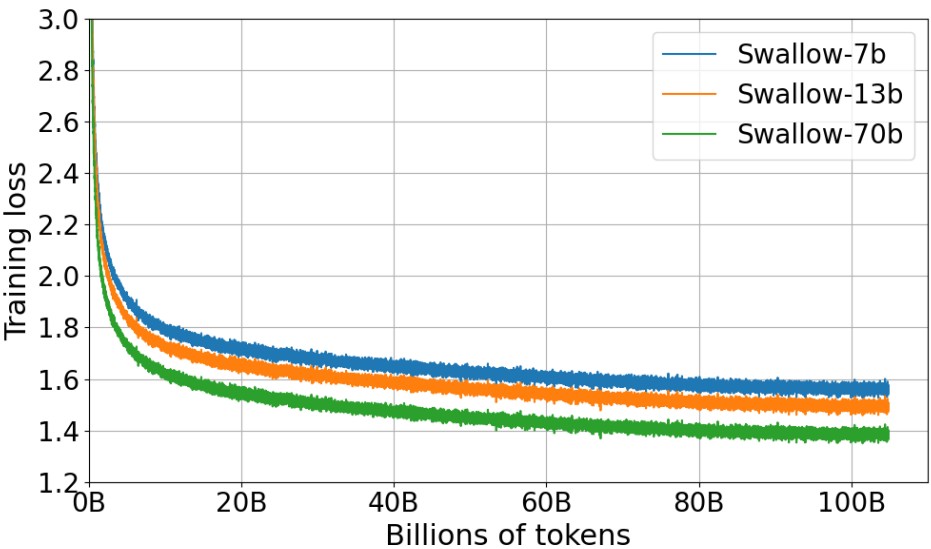

Figure 8: Training loss curves (number of tokens trained and loss values) for `Swallow` 7b, 13b, and 70b

## D    Distributed parallel training of models

Training large language models on a single GPU is challenging due to both GPU memory constraints and the time required for training. In terms of GPU memory, even using the latest H100 80GB, it is difficult to train the 7B model used in this study. Moreover, even if the model parameters, gradients, and optimizer states could fit on a single GPU, training on a single GPU would require an unrealistic amount of time to complete. Therefore, in this study, we adopted distributed parallel training, combining data parallelism and model parallelism.

### D.1    Training environment

We utilized the AI Bridging Cloud Infrastructure (ABCI) of the National Institute of Advanced Industrial Science and Technology, Japan for training. We employed mixed precision (`bfloat16`) and used multiple NVIDIA A100 nodes for distributed parallel training. Each node is equipped with eight NVIDIA A100 40GB GPUs, and the nodes are interconnected via InfiniBand HDR.

## D.2 Distributed training

| Number of Parameters | DP | TP | PP | SP | Distributed Optimizer |
|---|---|---|---|---|---|
| 7B | 16 | 2 | 2 | ✓ | ✓ |
| 13B | 8 | 2 | 4 | ✓ | ✓ |
| 70B | 4 | 8 | 8 | ✓ | ✓ |

Table 7: Distributed training settings. DP, TP, PP, and SP represent Data Parallelism, Tensor Parallelism, Pipeline Parallelism, and Sequence Parallelism, respectively.

To efficiently perform the training process, we adopted 3D parallelism, which integrates data parallelism, tensor parallelism, and pipeline parallelism, aiming for high computational efficiency and efficient memory utilization. We used the Megatron-LM[10] library for training. Table 7 shows the distributed training settings for each model size. In addition, we incorporated the following techniques:

**Efficient Memory Consumption**  By using the Distributed Optimizer in Megatron-LM, we distributed the optimizer states across data-parallel processes and eliminated redundancy, reducing the required memory usage. The Distributed Optimizer efficiently communicates using Reduce Scatter and All Gather, enabling memory reduction with the same communication cost as regular data parallelism.

**Topology-aware 3D Mapping**  In 3D parallelism, Transformer blocks are distributed across multiple GPUs using pipeline parallelism, and the parameters within each layer are distributed using tensor parallelism. As proposed in Megatron-LM (Narayanan et al., 2021), we placed the workers requiring more communication (tensor parallel workers) within the same node. This is because intra-node communication using NVLink is faster than inter-node communication. Additionally, considering the communication for gradient averaging in data parallelism, we placed data-parallel workers within the same node as much as possible. Pipeline parallelism requires less communication compared to other parallelization methods, using P2P (Point-to-Point) communication. Therefore, we placed pipeline stages across nodes.

**Adoption of 1F1B for Memory Efficiency**  By using PipeDream-Flush (Narayanan et al., 2021), a 1F1B (one forward pass followed by one backward pass) pipeline parallelism, we ensured that only a number of micro-batches less than or equal to the number of pipeline stages require activation, improving memory efficiency compared to GPipe (Huang et al., 2019).

**Parallelization using Sequence Parallelism**  Tensor parallelism parallelizes Self-Attention and MLP blocks, but not Layer-Norms and Dropouts, resulting in redundant memory usage of these components across tensor-parallel processes. To improve efficiency, we utilized Sequence Parallelism (Korthikanti et al., 2023). By using Sequence Parallelism in conjunction with tensor parallelism, memory efficiency can be achieved without the overhead of communication costs.

## D.3 Computational efficiency

Table 8 shows the computational performance in TFLOPS per GPU during the actual training process. In terms of execution efficiency, the 70B model achieved over 50%, indicating that the training was conducted efficiently[11].

---

[10]https://github.com/NVIDIA/Megatron-LM

[11]The theoretical values (312 TFLOPS) are extracted from the specifications of the NVIDIA A100 BFLOAT16 Tensor Core (without sparsity):
https://www.nvidia.com/en-us/data-center/a100/.

| # Params | # Nodes | TFLOPS/GPU | Execution efficiency |
|----------|---------|------------|---------------------|
| 7B | 4 | 134 | 43.0 % |
| 13B | 8 | 143 | 45.8 % |
| 70B | 32 | 158 | 50.6 % |

Table 8: TFLOPS/GPU for each model parameter

### D.4 Computation required for training

The training of `Swallow` 7b, 13b, and 70b required approximately $5.0 \times 10^{21}$ FLOPS, $9.4 \times 10^{21}$ FLOPS, $5.0 \times 10^{22}$ FLOPS of computation, respectively. [12]

## E Details of experimental setup

### E.1 Vocabulary expansion method

The Japanese vocabulary was constructed using a randomly sampled subset (1.5B tokens) of the word-segmented Swallow Corpus. However, to treat symbols as independent subwords, words containing symbols were split at both ends of the symbols. Following `Llama 2`, we used the BPE algorithm implemented in SentencePiece (Kudo & Richardson, 2018). Two post-processing steps were applied to the subword vocabulary constructed by SentencePiece. Firstly, the whitespace escape characters added by SentencePiece were removed. This is because, unlike during the vocabulary construction phase, word segmentation using MeCab is not performed in tokenization during training and inference. Secondly, the size of the additional vocabulary was intentionally adjusted to be a multiple of 8. This is a measure to facilitate distributed parallel training of the model.

The scores of the added subwords were used as-is from the BPE results output by SentencePiece. This decision was made based on the judgment that conflicts between the merge rules of the original vocabulary and the added vocabulary are extremely rare. In fact, the original vocabulary of the `Llama 2` tokenizer does not contain Japanese subwords of two or more characters.

## F Issues in evaluating natural language inference tasks

For the natural language inference datasets included in llm-jp-eval, more specifically, Jamp (Sugimoto et al., 2023), JaNLI (Yanaka & Mineshima, 2021), JNLI (Kurihara et al., 2022), JSeM (Kawazoe et al., 2015), and JSICK (Yanaka & Mineshima, 2022), we confirmed score fluctuations due to the class imbalance in multiple models. First, Figure 9 shows the class distributions of both the ground truth and the predictions made by `Swallow-7b`. Both the ground truth and predictions are greatly imbalanced, with the most frequent class accounting for over 95% of all predictions in three datasets. As a result, scores significantly fluctuate depending on whether the most frequent classes in predictions and ground truth align by chance. Figure 10 shows the learning curve of `Swallow-7b` on these datasets. While fluctuations of about 40 points occurred in two datasets, this was due to transitions in the most frequent class of predictions. The imbalance in prediction was not unique to `Swallow` but was also observed in other 7b and 13b models (Figure 11). From these observations, we concluded that it is misleading to discuss natural language inference abilities based solely on scores, leading to the exclusion from our evaluation. For a fair and stable evaluation of this task, addressing the imbalance in ground truth classes is recommended.

---

[12]The calculation of FLOPs is based on the formula presented in APPENDIX: FLOATING-POINT OPERATIONS of (Narayanan et al., 2021), with modifications made to adapt to the Llama architecture.

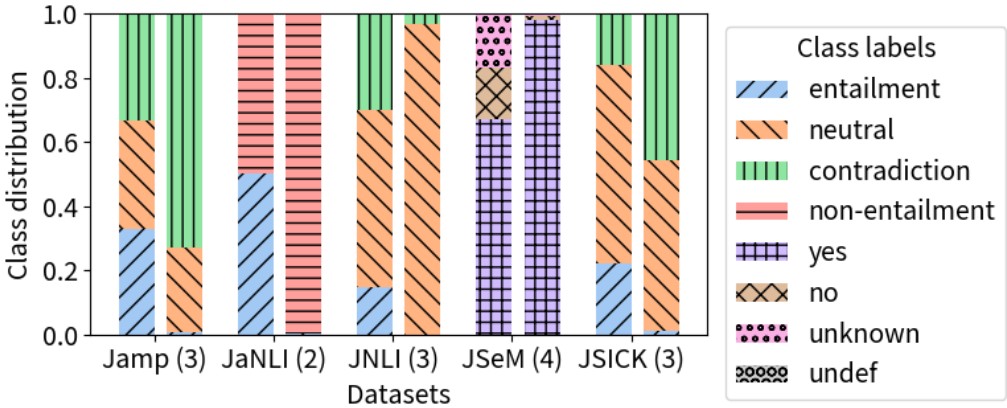

Figure 9: Class distributions in natural language inference task datasets. For each dataset, the left bar presents the ground truth, and the right bar does the prediction by `Swallow-7b`. The numbers in parentheses indicate the number of classes.

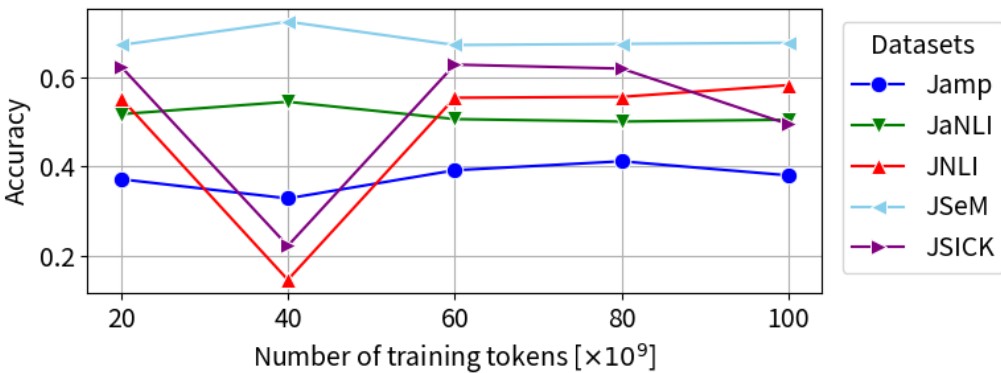

Figure 10: Accuracy curve of `Swallow-7b` on natural language inference during training.

## G   Evaluation methods for Japanese and English

Tables 2 and 3 present the evaluation benchmarks for Japanese and English, respectively.

The Japanese benchmarks include llm-jp-eval (Han et al., 2024) for a comprehensive evaluation of language models in Japanese, and the JP Language Model Evaluation Harness, which is designed for a more focused assessment of language model capabilities. The datasets cover a range of tasks including multiple-choice and open-ended question answering (JCQA (Kurihara et al., 2022), JEMHQA (Ishii et al., 2023), NIILC (Sekine, 2003)), machine reading comprehension (JSQuAD (Kurihara et al., 2022)), automatic summarization (XL-Sum (Hasan et al., 2021)), arithmetic reasoning (MGSM (Shi et al., 2023)), and machine translation (WMT'20 En-Ja and Ja-En (Barrault et al., 2020)). Below, we describe six notable task datasets included in these benchmarks.

**JCQA**   JCommonsenseQA (JCQA) is a Japanese question-answering dataset in a five-choice format that assesses common sense knowledge. It is the Japanese version of CommonsenseQA (Talmor et al., 2019), which was created to evaluate common sense reasoning ability, and consists of pairs of question sentence and answer choices. The knowledge covered by JCQA has a good balance of Japanese linguistic knowledge and common sense

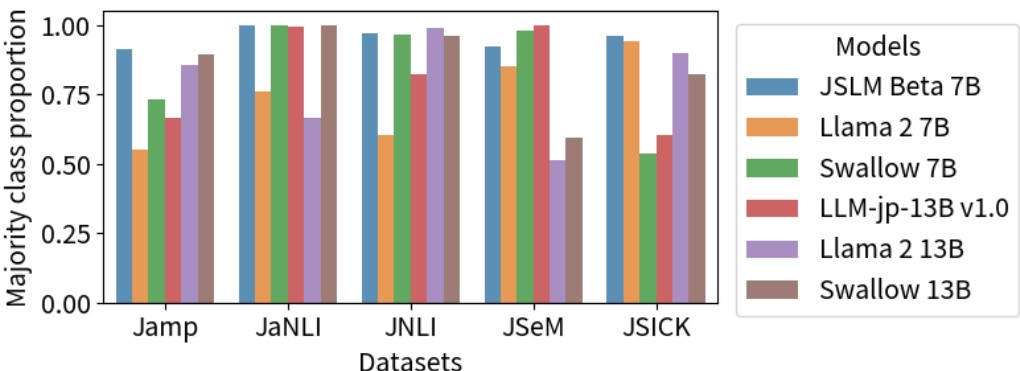

Figure 11: The proportion of the majority class in predictions by various LLMs.

knowledge. Moreover, since the incorrect answers are generated by language models, the dataset contains questions that cannot be solved just by avoiding semantically unrelated choices.

**JEMHQA** JEMHopQA (JEMHQA) is a Japanese open-ended question answering dataset that was originally designed as a multi-hop QA task. In our task setting, the dataset is used to evaluate a model's ability to generate answers directly from the given questions. The questions in JEMHQA are created using information from the Japanese version of Wikipedia, ensuring a diverse range of topics and entities. JEMHQA serves as an important benchmark for evaluating the extent of knowledge and ability to answer questions through reasoning with that knowledge.

**JSQuAD** JSQuAD is a Japanese machine reading comprehension dataset, which is the Japanese version of SQuAD (Rajpurkar et al., 2016). It focuses on question-answering style, which involves reading a document and a question, and then extracting a segment of text in the document as the answer. JSQuAD utilizes article paragraphs from the Japanese version of Wikipedia as its source documents.

**NIILC** NIILC is a dataset created for the purpose of developing question-answering systems in Japanese, featuring relatively straightforward questions that can be answered by referencing an encyclopedia. This makes it valuable for evaluating the encyclopedic knowledge of LLMs. We use only the question and answer pairs from the dataset, although the original dataset includes additional metadata like question types and evidence sentences.

**XL-Sum** XL-Sum Japanese version is a dataset created by extracting the Japanese portion from XL-Sum, a large-scale summarization dataset collected from BBC News articles, and further filtering it into a subset suitable for abstractive summarization. The dataset was filtered by calculating the 15-gram overlap rate between articles and summaries, selecting pairs with low overlap rates. Consequently, this dataset necessitates the ability to paraphrase and abstract information, rather than simply extracting sentences.

**MGSM** MGSM is a dataset created by selecting 250 elementary school arithmetic word problems from GSM8K (Cobbe et al., 2021) and manually translating them. As it deals with Japanese arithmetic word problems, it requires both Japanese linguistic knowledge and mathematical reasoning ability.

| | | | Evaluation in Japanese | | | | | | | |
|---|---|---|---|---|---|---|---|---|---|---|
| Model | Tokens | JCQA | JEMHQA | NIILC | JSQuAD | XL-Sum | MGSM | En-Ja | Ja-En | Avg |
| Swallow-7B-VE | 20 | 39.2 | 55.4 | 51.2 | 83.5 | 14.7 | 7.2 | 22.4 | 14.6 | 36.0 |
| | 40 | 45.8 | 52.4 | 52.6 | 84.6 | 15.9 | 8.8 | 23.6 | 14.4 | 37.2 |
| | 60 | 42.2 | 51.2 | 58.3 | 84.3 | 18.4 | 10.4 | 24.6 | 14.7 | 38.0 |
| | 80 | 49.2 | 49.5 | 60.3 | 86.0 | 18.4 | 9.6 | 24.9 | 15.0 | 39.1 |
| | 100 | 48.1 | 50.8 | 59.7 | 85.7 | 18.3 | 12.4 | 25.1 | 15.1 | 39.4 |
| Swallow-7B-¬VE | 20 | 45.8 | 52.2 | 48.7 | 84.6 | 19.0 | 8.0 | 21.8 | 13.6 | 36.7 |
| | 40 | 39.1 | 54.1 | 52.7 | 84.7 | 18.2 | 7.2 | 22.7 | 14.7 | 36.7 |
| | 60 | 49.2 | 54.9 | 57.2 | 86.4 | 20.4 | 9.2 | 23.6 | 15.2 | 39.5 |
| | 80 | 52.3 | 55.4 | 58.6 | 87.1 | 21.8 | 8.8 | 23.8 | 15.1 | 40.4 |
| | 100 | 54.3 | 54.3 | 57.3 | 86.8 | 21.2 | 12.0 | 24.1 | 15.1 | 40.6 |
| Swallow-13B-VE | 20 | 76.6 | 51.8 | 58.6 | 88.5 | 18.9 | 12.0 | 24.4 | 16.0 | 43.3 |
| | 40 | 75.3 | 52.0 | 59.4 | 88.8 | 19.5 | 15.6 | 25.9 | 16.3 | 44.1 |
| | 60 | 71.9 | 50.5 | 63.1 | 89.4 | 20.6 | 18.0 | 27.0 | 17.3 | 44.7 |
| | 80 | 75.8 | 48.9 | 63.5 | 89.7 | 21.2 | 22.8 | 27.1 | 17.5 | 45.8 |
| | 100 | 78.4 | 50.6 | 64.0 | 90.1 | 21.7 | 20.4 | 27.2 | 17.7 | 46.3 |
| Swallow-13B-¬VE | 20 | 72.9 | 46.7 | 54.0 | 88.6 | 22.7 | 12.0 | 23.7 | 14.5 | 41.9 |
| | 40 | 69.4 | 50.5 | 60.0 | 89.2 | 21.0 | 12.8 | 25.8 | 16.8 | 43.2 |
| | 60 | 76.4 | 53.9 | 61.6 | 89.9 | 23.1 | 16.0 | 26.7 | 17.4 | 45.6 |
| | 80 | 76.9 | 51.4 | 63.8 | 90.0 | 23.0 | 23.2 | 26.8 | 17.3 | 46.6 |
| | 100 | 77.1 | 54.4 | 63.5 | 90.3 | 22.9 | 21.2 | 27.4 | 18.2 | 46.9 |
| Swallow-70B-VE | 20 | 90.8 | 59.9 | 65.3 | 92.0 | 22.2 | 38.0 | 28.4 | 22.3 | 52.4 |
| | 40 | 92.1 | 62.0 | 69.2 | 92.0 | 22.6 | 40.0 | 29.8 | 22.1 | 53.7 |
| | 60 | 92.2 | 61.2 | 67.8 | 91.8 | 22.7 | 48.0 | 29.7 | 22.7 | 54.5 |
| | 80 | 93.4 | 60.4 | 69.6 | 91.8 | 22.5 | 47.2 | 30.6 | 23.1 | 54.8 |
| | 100 | 93.5 | 62.9 | 69.6 | 91.8 | 22.7 | 48.4 | 30.4 | 23.0 | 55.3 |
| Swallow-70B-¬VE | 20 | 91.3 | 57.1 | 63.1 | 92.2 | 26.2 | 38.0 | 28.7 | 22.2 | 52.4 |
| | 40 | 92.8 | 54.6 | 64.5 | 91.8 | 24.7 | 40.8 | 29.8 | 23.0 | 52.7 |
| | 60 | 93.5 | 58.1 | 68.5 | 92.2 | 26.4 | 42.8 | 30.0 | 23.0 | 54.3 |
| | 80 | 93.9 | 58.6 | 70.0 | 92.3 | 27.4 | 48.4 | 30.2 | 22.6 | 55.4 |
| | 100 | 94.1 | 57.6 | 70.2 | 92.5 | 27.6 | 47.2 | 30.4 | 23.2 | 55.4 |

Table 9: Evaluation of `Swallow` models in Japanese tasks with and without vocabulary expansion. "Tokens" represent the number of training tokens.

## H  Detail analysis of vocabulary expansion impact

Table 9 and 10 show the absolute performance scores for `Swallow` with and without vocabulary expansion (`Swallow` ¬VE). While Figure 4 shows relative changes, this table provides a comprehensive view of the absolute scores across different evaluation tasks.

Regarding Japanese language capabilities, we analyzed the overall impact of vocabulary expansion on performance across various tasks and model sizes. Excluding the XL-Sum task, the only dataset where we observed a relative change of more than 10% is JCQA for the `Swallow-7B-VE` model. This kind of change was not observed in other model sizes or evaluation tasks. Table 9 indicates that the training trajectory of JCQA score are not stable. Unlike the XL-Sum task, where the model without vocabulary expansion (¬VE) consistently outperforms the model with vocabulary expansion (VE), JCQA does not exhibit such a consistent trend. For instance, at 40B training tokens, the VE model outperforms the ¬VE model in JCQA. Based on these evaluation results and the observed instability, we conclude that the impact of vocabulary expansion on overall performance is minor except for XL-Sum task.

| Model | Tokens | Evaluation in English | | | | | | |
|---|---|---|---|---|---|---|---|---|
| | | OBQA | TrQA | HS | SQuAD2 | XW | GSM8K | Avg |
| Swallow-7B-VE | 20 | 29.8 | 46.5 | 53.1 | 28.6 | 88.6 | 7.8 | 42.4 |
| | 40 | 31.8 | 43.8 | 52.3 | 30.5 | 88.2 | 7.7 | 42.4 |
| | 60 | 31.2 | 46.1 | 52.4 | 31.3 | 88.3 | 9.2 | 43.1 |
| | 80 | 31.4 | 47.9 | 52.7 | 30.4 | 88.7 | 9.7 | 43.5 |
| | 100 | 31.8 | 48.4 | 53.1 | 31.3 | 88.2 | 11.3 | 44.0 |
| Swallow-7B-¬VE | 20 | 30.6 | 48.2 | 52.9 | 27.6 | 87.8 | 7.9 | 42.5 |
| | 40 | 30.6 | 46.2 | 52.2 | 29.2 | 88.0 | 8.0 | 42.4 |
| | 60 | 30.0 | 48.6 | 52.9 | 31.2 | 88.7 | 9.7 | 43.5 |
| | 80 | 32.0 | 49.6 | 53.0 | 29.8 | 88.5 | 8.5 | 43.6 |
| | 100 | 31.8 | 50.8 | 53.3 | 29.2 | 88.2 | 9.9 | 43.9 |
| Swallow-13B-VE | 20 | 33.6 | 56.8 | 56.3 | 30.5 | 89.6 | 14.8 | 46.9 |
| | 40 | 33.0 | 53.4 | 55.5 | 33.4 | 89.6 | 15.9 | 46.8 |
| | 60 | 34.0 | 55.5 | 55.7 | 32.4 | 89.9 | 17.4 | 47.5 |
| | 80 | 33.8 | 58.2 | 56.2 | 33.0 | 90.3 | 19.8 | 48.5 |
| | 100 | 35.0 | 58.5 | 56.6 | 34.1 | 90.8 | 20.4 | 49.2 |
| Swallow-13B-¬VE | 20 | 35.2 | 56.8 | 56.2 | 31.1 | 89.9 | 13.7 | 47.1 |
| | 40 | 33.6 | 55.2 | 55.6 | 33.4 | 89.9 | 15.6 | 47.2 |
| | 60 | 33.4 | 57.5 | 56.7 | 34.5 | 89.9 | 16.8 | 48.1 |
| | 80 | 35.2 | 59.3 | 56.7 | 34.5 | 90.0 | 17.0 | 48.8 |
| | 100 | 34.6 | 60.3 | 57.0 | 34.8 | 90.1 | 17.5 | 49.0 |
| Swallow-70B-VE | 20 | 41.6 | 78.2 | 64.1 | 32.7 | 92.4 | 39.9 | 58.1 |
| | 40 | 41.0 | 76.0 | 64.1 | 35.9 | 92.2 | 43.5 | 58.8 |
| | 60 | 42.2 | 76.8 | 64.2 | 36.3 | 92.1 | 45.4 | 59.5 |
| | 80 | 42.6 | 77.2 | 64.4 | 37.4 | 92.2 | 48.3 | 60.4 |
| | 100 | 42.2 | 77.6 | 64.6 | 37.5 | 92.0 | 48.7 | 60.4 |
| Swallow-70B-¬VE | 20 | 40.8 | 78.1 | 64.2 | 35.1 | 92.3 | 41.1 | 58.6 |
| | 40 | 41.6 | 76.8 | 64.0 | 31.0 | 91.9 | 43.4 | 58.1 |
| | 60 | 41.6 | 77.1 | 64.2 | 34.0 | 92.1 | 46.9 | 59.3 |
| | 80 | 43.6 | 78.0 | 64.6 | 33.7 | 92.4 | 49.1 | 60.2 |
| | 100 | 42.0 | 78.3 | 64.7 | 33.3 | 92.2 | 49.7 | 60.0 |

Table 10: Evaluation of `Swallow` models in English tasks with and without vocabulary expansion. "Tokens" represent the number of training tokens.

# I Comparison of models trained from scratch and other continual pre-training models

Table 11 shows the evaluation results of pre-trained models and continual pre-training models, with the majority developed in Japan (see Table 12 in the appendix for the sources of the models). Some LLMs trained from scratch outside of Japan (Mistral v0.1, Qwen-7B, Qwen-14B) show higher Japanese performance than Llama 2. The models that are continually pre-trained from these LLMs (japanese-stablelm-base-gamma-7b, nekomata-7b, nekomata-14b) show higher average scores than `Swallow`, suggesting that the performance differences in the base models are reflected.

## I.1 Details of models trained from scratch

**calm2-7b** A model with the Llama architecture trained on 1.3T tokens of Japanese and English corpora.

**llm-jp-13b-v1.0** A model with the GPT-2 architecture trained on 300B tokens of Japanese and English corpora.

**PLaMo-13b** A model trained on 180B tokens of Japanese, 1.32T tokens of English, totaling 1.5T tokens.

| Model | Japanese evaluation dataset | | | | | | | | |
|---|---|---|---|---|---|---|---|---|---|
| | JCQA | JEMHQA | NIILC | JSQuAD | XL-Sum | MGSM | En-Ja | Ja-En | Avg |
| calm2-7b | 22.0 | 50.5 | 50.7 | 78.0 | 2.3 | 6.0 | 23.5 | 15.0 | 31.0 |
| J Stable LM $\beta$ 7b | 36.1 | 44.8 | 44.3 | 83.2 | 22.0 | 7.2 | 19.5 | 12.3 | 33.7 |
| ELYZA-7b | 57.9 | 47.0 | 40.2 | 82.3 | 13.1 | 6.0 | 18.0 | 12.9 | 34.7 |
| youri-7b | 46.2 | 47.8 | 50.0 | 85.1 | 19.6 | 6.4 | 26.7 | **19.7** | 37.7 |
| Mistral v0.1 7B | 73.0 | 42.5 | 27.2 | 85.6 | 20.1 | 17.6 | 14.1 | 17.3 | 37.2 |
| J Stable LM $\gamma$ 7b | 73.6 | 46.4 | 55.7 | **89.1** | **22.9** | 16.8 | 23.9 | 15.6 | **43.0** |
| Qwen-7B | **77.1** | 42.3 | 23.8 | 85.9 | 13.7 | **21.6** | 16.9 | 18.0 | 37.4 |
| nekomata-7b | 74.2 | 49.3 | 50.2 | 87.1 | 16.8 | 12.4 | **26.7** | 18.2 | 41.8 |
| Llama-2-7b | 38.5 | 42.4 | 34.1 | 79.2 | 19.1 | 7.6 | 17.8 | 17.4 | 32.0 |
| Swallow-7b | 48.1 | **50.8** | **59.7** | 85.7 | 18.3 | 12.4 | 25.1 | 15.1 | 39.4 |
| llm-jp-13b-v1.0 | 22.6 | 47.9 | 38.6 | 77.4 | 10.8 | 2.4 | 19.6 | 11.9 | 28.9 |
| PLaMo-13b | 22.7 | 51.9 | 41.4 | 76.2 | 10.3 | 3.6 | 15.8 | 12.0 | 29.2 |
| ELYZA-13b | 74.0 | 42.6 | 46.8 | 87.2 | 14.1 | 3.2 | 22.0 | 14.9 | 38.1 |
| Qwen-14B | 88.3 | 42.4 | 32.2 | 89.8 | 18.5 | **38.8** | 22.2 | 22.2 | 44.3 |
| nekomata-14b | **91.7** | **57.8** | 61.1 | **91.5** | 21.3 | 35.6 | **29.9** | **23.1** | **51.5** |
| Llama-2-13b | 70.0 | 44.2 | 41.7 | 85.3 | 21.4 | 13.2 | 21.5 | 19.8 | 39.6 |
| Swallow-13b | 78.4 | 50.6 | **64.0** | 90.1 | **21.7** | 20.4 | 27.2 | 17.7 | 46.3 |
| J Stable LM $\beta$ 70b | 91.2 | 49.3 | 60.4 | **91.9** | **25.7** | 41.6 | 27.7 | 23.4 | 51.4 |
| Llama-2-70b | 86.9 | 46.6 | 52.6 | 90.8 | 23.6 | 35.6 | 26.4 | **24.0** | 48.3 |
| Swallow-70b | **93.5** | **62.9** | **69.6** | 91.8 | 22.7 | **48.4** | **30.4** | 23.0 | **55.3** |

Table 11: Evaluation results on Japanese evaluation datasets.

## J  Sources of evaluated models

Table 12 lists the names of the models used in the experiments and their corresponding Hugging Face Model Names. The abbreviations in the "Training detail" column of Table 12 are defined as follows: VE for vocabulary expansion, CT for continual pre-training, and PT for pre-training from scratch.

The models with "Ja & En" in their Training detail were trained on both Japanese and English data, while those with "Zh & En" were trained using Chinese and English data. Models with "Ja" were trained solely on Japanese data, and those with "En" were primarily trained on English data.

Note that the "Training detail" column presents the major data source used for training. However, some models may also use small amounts of other types of data in their training data, for example, source code, mathematical texts, and corpora in other languages.

| Model | Training detail | Model name on Hugging Face |
|---|---|---|
| calm2-7b | (PT, Ja & En) | cyberagent/calm2-7b |
| J Stable LM $\beta$ 7b | (CT, Ja & En) | stabilityai/japanese-stablelm-base-beta-7b |
| ELYZA-7b | (CT, Ja) | elyza/ELYZA-japanese-Llama-2-7b |
| youri-7b | (CT, Ja & En) | rinna/youri-7b |
| Mistral v0.1 (7B) | (PT, En) | mistralai/Mistral-7B-v0.1 |
| J Stable LM $\gamma$ 7b | (CT, Ja & En) | stabilityai/japanese-stablelm-base-gamma-7b |
| Qwen-7B | (PT, Zh & En) | Qwen/Qwen-7B |
| nekomata-7b | (CT, Ja & En) | rinna/nekomata-7b |
| Llama-2-7b | (PT, En) | meta-llama/Llama-2-7b-hf |
| Swallow-7b | (CT, VE, Ja & En) | tokyotech-llm/Swallow-7b-hf |
| llm-jp-13b-v1.0 | (PT, Ja & En) | llm-jp/llm-jp-13b-v1.0 |
| PLaMo-13b | (PT, Ja & En) | pfnet/plamo-13b |
| ELYZA-13b | (CT, Ja) | elyza/ELYZA-japanese-Llama-2-13b |
| Qwen-14B | (PT, Zh & En) | Qwen/Qwen-14B |
| nekomata-14b | (CT, Ja & En) | rinna/nekomata-14b |
| Llama-2-13b | (PT, En) | meta-llama/Llama-2-13b-hf |
| Swallow-13b | (CT, VE, Ja & En) | tokyotech-llm/Swallow-3b-hf |
| J Stable LM $\beta$ 70b | (CT, Ja & En) | stabilityai/japanese-stablelm-base-beta-70b |
| Llama-2-70b | (PT, En) | meta-llama/Llama-2-70b-hf |
| Swallow-70b | (CT, VE, Ja & En) | tokyotech-llm/Swallow-70b-hf |

Table 12: Evaluated models and their distribution URLs

