# OpenReview forum: "Continual Pre-Training for Cross-Lingual LLM Adaptation: Enhancing Japanese Language Capabilities"
_colmweb.org/COLM/2024/Conference — COLM_

### Official Review · Reviewer_Uybi · 2024-04-26

**Rating:** 5
**Confidence:** 4
**Ethics Flag:** 1

**Summary:**

This paper presents an effort in continuing pre-training an LLM to improve Japanese language capabilities. Authors applied and analysed vocabulary expansion and parallel corpus. Finally llama-2 at 7B, 30B, and 70B are continue-pre-trained with more text in Japanese. Through language-oriented tests, it is found that Japanese test scores mostly increase, but English scores drop. These models achieve better scores in comparison with LLMs trained from scratch in Japanese.

**Questions To Authors:**

I would like to clarify with the authors some points below which I think can affect the soundness of the work:

1. Abstract: "*The results showed that the efficiency gained through vocabulary expansion had no negative impact on performance*". I wonder what "*efficiency*" means here?
2. Sec 2.2: "*For example, a Japanese single-character noun 猫 (cat) is represented by three byte-level tokens <0xE7> <0x8C> <0xAB>, which do not provide any semantic meaning*"
    - I understand that if a string is not in the original tokenizer, it will be split up in a more fine-grained way.
    - However, I might not agree with the statement because it is discussed from the perspective of human readers.
    - If the three byte-level tokens appear together and often enough during LLM training, they do provide semantic information when being encoded by the model. In other words, I do not see why "猫 (cat)" provides more semantic meaning.
3. Sec 3.3: As a research paper, I'd ask about the reason behind the decision of a Japanese vocabulary of "*(up to 16k)*", and why the new embeddings are "*initialized with the average of the vectors of the subwords segmented by the LLaMA tokenizer*".
4. Sec 5.1.2 and Figure 4: Is "Llama 2-JA without VE" the reference point, and do you have individual absolute scores for it? I also find the statement of "*overall impact of vocabulary expansion on performance is minor.*" to be a bit hand waving because for some tasks and gain/loss are more than 10%, translating to significant absolute differences in test scores.

**Reasons To Accept:**

- the technical points are detailed and easy-to-follow. This work can be consulted by other researchers and engineers looking to continual pre-train a base LLM.
- Without any doubt, if released, the trained checkpoints will be adopted by practitioners to deal with Japanese texts.
- the incorporation of parallel corpus into continued pre-training is novel. Note that I have seen it somewhere else [1], but I agree, that both works are rather concurrent.

[1] TowerLLM; https://arxiv.org/abs/2402.17733

**Reasons To Reject:**

- I find this work lacks novelty and depth from a research perspective. Continued pre-training has been done and reported by some researchers; the technique of vocabulary expansion has been widely used. I did do not find many new insights in this paper.
- It is observed that English test scores do drop after continued pre-training, and the paper does not discuss or propose any techniques to mitigate the issues. Is it possible to do continued pre-training with both English and Japanese as opposed to the current setting?
- There are some confusions or concerns about soundness of the arguments or claims, which I detail in the questions section below.

---

> ### Author Rebuttal · Authors · 2024-05-29
>
> Thank you for your review.
>
> >  lacks novelty and depth
>
> We confirmed the training scalability and effectiveness of continual pre-training across different model sizes. This fact has not been investigated comprehensively in the previous studies (§2.1). Our analysis of vocabulary expansion (VE) in §5.1.2 showed that the effect of VE varies across different tasks, which was a non-trivial finding.
>
> Additionally, we released SoTA Japanese LLMs (as of submission), which also make a non-trivial finding for developing Japanese foundation models. These contributions align with the "LM for everyone" topic in the CFP.
>
> > English test scores do drop
>
> Teaching new facts to pre-trained LLMs can degrade the original capability [Gudibande et al., 2024; Gekhman et al., 2024]. Therefore, inspired by experience replay [Chaudhry et al., 2019], we took measures to mitigate the negative impact on English by using a multilingual corpus of Japanese and English (§3.2). Table 4 shows the effectiveness of these mitigations, especially for larger models (70B). In addition, we want to emphasize that the primary goal of this study is to develop a Japanese LLM (§1).
>
> > what "efficiency" means here?
>
> "Efficiency" refers to the reduction of the number of tokens required to encode/decode the same text.
>
> > "猫 (cat)"
>
> Your argument may be true for computers. However, the three byte-level tokens do not correspond to the morphology, phonology, and semantics of 猫.
>
> > Japanese vocabulary size "up to 16k"
>
> A preliminary experiment (§E.1) showed that the sequence lengths with 16k, 32k, and 48k Japanese vocabularies were similar (43.8%, 45.9%, and 46.8%, respectively, of that with the original vocabulary), and concluded that 16k was sufficient. We will include this in §3.3 of the camera-ready version.
>
> > new embeddings are "initialized with the average"
>
> New embeddings were not trained at all compared to the existing ones. We followed Yao et al. (2021) (§3.3). Additionally, we confirmed that random initialization caused corrupted generation. We will include this in §3.3.
>
> > Sec 5.1.2 and Figure 4
>
> Correct. Figure 4 shows the relative change using "Llama 2-JA without VE" as the reference point. We will add the absolute score changes to the appendix.
>
> > "impact of vocabulary expansion on performance is minor."
>
> Among the Japanese evaluation tasks, except for XL-Sum, we described the results as "minor" due to the inconsistent positive and negative impacts across different model sizes (§5.1.2).

---

> > ### Comment · Reviewer_Uybi · 2024-06-02
> > **Reviewer Uybi's response after rebuttal**
> >
> > Thank you to the authors for the explanations and clarifications. I re-read the paper while referencing to your rebuttal and related works. I feel that I have a better understanding of the contribution of this paper now. While most points are understood, I feel that I still have reserved opinions about the two statement below.
> >
> > **1.** Re: intact token v. byte-level tokens: whether the former provides more semantics when fed to an LLM.
> >
> > The original statement and the explanation are both centred around human interpretation of words/characters. After rebuttal, I feel unsure about whether/how "morphology" or "phonology" are used by an LLM if the tokens are mapped to indices and then vectors. LLM's embeddings (so-called semantics) are derived by contextualizing on the input itself and surrounding tokens. Note that I completely agree with the choice of expanding the vocabulary for extra language support. It improves *tokenizer fertility*. I find the current justification (relying on semantics) unsupported.
> >
> > **2.** I feel that the statement "impact of vocabulary expansion on performance is minor" is not accurate.
> >
> > I would say that if some changes are noticeably positive while some are negative (10% relative change is a lot), it actually means that the impact on test performance is significant rather than "minor"?

---

> > > ### Author Response · Authors · 2024-06-04
> > >
> > > >  I find the current justification (relying on semantics) unsupported.
> > >
> > > We appreciate the reviewer's insights and understanding of the concerns regarding the use of "morphology" and "phonology" in the context of LLMs where tokens are mapped to indices and vectors. Our justification for stating that byte-level tokens "do not provide any semantic meaning" is based on three primary reasons: semantic integrity, prevention of generating non-existent characters, and improved tokenization efficiency and contextualization.
> > >
> > >
> > > 1. Semantic Integrity:
> > > Byte-level tokens can result in shared tokens for characters with vastly different meanings. For instance, "猟" (hunt) is encoded as <0xE7> <0x8C> <0x9F>, and "猫" (cat) is encoded as <0xE7> <0x8C> <0xAB>, sharing with the identical first two bytes despite their different meanings. This overlap can dilute the semantic integrity of Japanese kanji letters, which are ideograms unlike most of other languages. It is possible to argue that English also suffers from the same problem, e.g., “cat” and “car” are totally different, but the treatment of English tokenization is much better than that of Japanese because the tokenizer of the base model was trained on a corpus with a lot of English text and little Japanese text. In other words, it is unlikely for the base tokenizer to split them into “cat” and “car” into <c> <a> <t> and <c> <a> <r>. While LLMs derive semantics from context, starting with more semantic Japanese tokens is reasonable because Japanese kanji letter are ideograms.
> > >
> > >
> > > 2. Prevention of Generating Non-existent Characters:
> > > Byte-level tokenization can produce byte sequences that do not correspond to any valid characters in the target language. For instance, [it is known](https://okumuralab.org/~okumura/misc/230611.html) in Japan that ChatGPT generated the non-existent Japanese word "視覴," allegedly caused by the impossible combination of byte tokens from "覚(perception)" and "聴(listen)." Such occurrences are less likely if we treat a Japanese kanji as a single token, which respects the boundaries of valid characters and words. This reduces the possibility of generating meaningless or incorrect text and thereby improves the reliability and quality of the model's output.
> > >
> > >
> > > 3. Improved Tokenization Efficiency and Contextualization:
> > > Representing a kanji letter as a single token provides a more consistent and semantically relevant input to the token embeddings, enhancing the model's ability to understand and generate coherent text. For example, "猫"(cat) = <0xE7> <0x8C> <0xAB> shares the first two tokens with 17 other kanji in byte-level tokenization, which potentially increases the semantic ambiguity and degrades its performance. By adding kanji letters to the vocabulary, we can ensure that each token has a clear and distinct meaning, which improves the model's ability to contextualize and generate accurate representations.
> > >
> > >
> > > >  I feel that the statement "impact of vocabulary expansion on performance is minor" is not accurate.
> > >
> > > Thank you for your feedback. we would like to clarify our position regarding the statement, "the impact of vocabulary expansion on performance is minor."
> > >
> > > Excluding the XL-Sum task, the only dataset where we observed a relative change of more than 10% is JCQA for the Llama-2-JA-7b model. This kind of change was not observed in other model sizes or tasks, as we noted in §5.1.2.
> > >
> > > To provide more evidence, we present an additional analysis of the JCQA scores, isolated from the averaged numbers reported in Figure 7. The scores for 7b models’ JCQA with and without vocabulary expansion (VE and $\neg$VE, respectively) are as follows:
> > > |Num. of training tokens (B)|JCQA(VE)|JCQA($\neg$VE)|
> > > |---|---|---|
> > > |0|38.52|38.52|
> > > |20|39.23|45.76|
> > > |40|45.76|39.14|
> > > |60|42.18|49.15|
> > > |80|49.15|52.28|
> > > |100|48.08|54.33|
> > > With 40B tokens used for training, the model with vocabulary expansion (VE) outperforms the one without it ($\neg$VE). This indicates that the JCQA scores are not stable. In contrast, Figure 6 shows that the XL-Sum scores consistently show better performance for the model without vocabulary expansion, exhibiting a different trend.
> > >
> > > Furthermore, the average and standard deviation of JCQA scores after training 40B tokens (40,60,80,100 B Tokens) are:
> > >
> > > - Vocabulary Expansion (VE): 46.29 ± 3.09
> > > - No Vocabulary Expansion ($\neg$VE): 48.73 ± 6.74
> > >
> > > The observed 6.25-point (11.5%) difference of the JCQA scores between the vocabulary expansion (VE) and non-vocabulary expansion ($\neg$VE) of the Llama-2-JA-7b models does not indicate a substantial impact because of the instability and large standard deviation on the dataset.
> > >
> > > Therefore, considering these evaluation results and the observed instability, we regard that the impact of vocabulary expansion on overall performance is minor.
> > >
> > > In the camera-ready version, we will include these elaborations in the Appendix.

---

### Official Review · Reviewer_ATKc · 2024-05-10

**Rating:** 6
**Confidence:** 4
**Ethics Flag:** 1

**Summary:**

This works presents a new version of LLAMA2, adapted for Japanese. First, LLAMA's vocabulary is expanded with a Japanese vocabulary, created from the new training data through MeCab, Unidic and BPE. Then, the model is trained by performing continual pre-training. Finally, the new model is tested on several downstream tasks, achieving satisfying results.

**Reasons To Accept:**

The work describes the construction of a new model and an analysis which could benefit the community.

**Reasons To Reject:**

As it is right now, the paper relies too much on the appendixes and is very hard to read and understand. Despite its importance, the experimental setup is missing. Similarly, the evaluation lacks context: It is mentioned which downstream task are being used for the evaluation, but there is no description even of which metrics and corpora are being used. Thus, result tables are not informative enough--we don't even know if greater values means better results or if the corpora could have already been seen at training time!

While I find the work interesting, I believe it would benefit from a revision. The main details should be described on the document's body instead of relaying so much on the appendixes.

---

> ### Author Rebuttal · Authors · 2024-05-28
>
> Thank you for your review and comments.
>
> > paper relies too much on the appendixes
>
> To meet the page limit, we carefully selected the content to include in the main text and the appendices. In the camera-ready version, we plan to add one more page, allowing us to move the vocabulary expansion method and the usage of parallel corpora into the main text, which should enhance readability and understanding.
>
> > Despite its importance, the experimental setup is missing.
>
> We have described our experimental setups in sections 3 of the manuscript. To provide an overview: we use Llama 2 as base models,  expand vocabulary for Japanese, and conduct continual pre-training using a multilingual corpus. To improve clarity, we will rename Section 3 to "Experimental settings of continual pre-training" and restructure Section 3.1 and 3.2 to clearly distinguish between the experimental settings and the motivations/rationale behind them.
>
> > the evaluation lacks context
>
> We would like to clarify that the evaluation tasks and metrics are detailed in Section 3.4 and Tables 2 and 3 of our paper. These tables specify the datasets used and the evaluation metrics. To further clarify, we use few-shot In-Context Learning to solve downstream tasks without fine-tuning, and we have not (intentionally) included evaluation data in the pre-training datasets.In the camera-ready version, we will explicitly state that higher scores indicate better results. Additionally, we will include more information on our approach of using few-shot In-Context Learning without fine-tuning and confirm that the evaluation datasets were not used in the pre-training process.

---

> > ### Author Response · Authors · 2024-06-04
> > **Initiating the discussion**
> >
> > We sincerely thank the reviewer for the valuable suggestions. We hope our response has thoroughly addressed all concerns and that the reviewer can consider improving their score based on our response. There is nothing we want to add to the response, but we are also happy to discuss with the reviewer if they still have a concern.

---

> > ### Comment · Reviewer_ATKc · 2024-06-04
> >
> > Thank you for your clarifications. I think that the extra page for the camera-ready will benefit the readability and understanding of the paper.

---

### Official Review · Reviewer_rjny · 2024-05-12

**Rating:** 7
**Confidence:** 4
**Ethics Flag:** 1

**Summary:**

The authors present work that extends baseline Llama-2 large language models to include (more) Japanese-language data through continued pre-training on up to 100B additional tokens of Japanese data. By employing model vocabulary expansion in conjunction with the additional data, the authors improve generation efficiency by up to 78% due to the introduction of Japanese-specific vocabulary terms. Experimental results show marked improvements on Japanese performance on six evaluation tasks in addition to improvements in English-Japanese machine translation (MT). Some performance degradation is noted in English-based evaluation tasks as well as Japanese-English MT performance. Lastly, the authors compare their continued pre-training approach against similarly-sized Japanese language models trained from scratch.

**Questions To Authors:**

It would be interesting to see results of other ways of mitigating 'catastrophic forgetting' such as Elastic Weight Consolidation in place of Experience Replay, though the implementation details may be tricky.

**Reasons To Accept:**

The contributions in this paper are systematically explored to show effects of the two major contributions. Vocabulary expansion is demonstrated to increase generation efficiency and generally has little effect on task evaluation (with the exception of a minor performance regression on summarization.) Continued training with additional Japanese monolingual and Japanese-English parallel data improves performance on Japanese evaluation tasks. The authors also demonstrate that pretraining with a strong English baseline model outperforms similar-sized models that were trained from scratch. This pretraining approach is tested on the three common Llama-2 model sizes 7B, 13B and 70B to show that this tactic scales with differing model sizes.

**Reasons To Reject:**

Language-specific LLM pretraining has been explored in many other works, but the combination of examination of model size and effects of vocabulary expansion may help mitigate this fact.

---

> ### Author Rebuttal · Authors · 2024-05-28
>
> Thank you for your review and comments.
>
> > Elastic Weight Consolidation
>
> We agree with your suggestion. Perhaps due to the capacity of the model, there tends to be a decline in the original English proficiency during continual pre-training. Indeed, mitigating this decline is crucial, and we have been implementing a strategy to prevent catastrophic forgetting. We also appreciate your suggestion to explore Elastic Weight Consolidation as another potential approach.
>
> > Language-specific LLM pretraining has been explored in many other works, but the combination of examination of model size and effects of vocabulary expansion may help mitigate this fact.
>
> We investigated the impact of vocabulary expansion comprehensively across various model sizes, along with the effectiveness of incorporating parallel corpora. We believe this is particularly beneficial for developing large language models for non-English speaking regions to enhance large language models for targeted languages.

---

> > ### Comment · Reviewer_rjny · 2024-06-05
> > **Acknowledgement of Rebuttal**
> >
> > Thank you for addressing my review comments - as discussed by other reviewers, the paper may benefit from some portions of the appendix moved into the main paper if afforded an extra page upon acceptance.

---

### Decision · Program_Chairs · 2024-07-10

**Decision:**

Accept

**Comment:**

This paper describes an approach to continual pre=train LLaMA for Japanese using 100 billion additional tokens.

Strengths:
1. a detailed description of the procedure, including vocabulary expansion, continual pre-training step.
2. Evaluation on a variety of model sizes show its effectiveness.

Weakness:
1. Limited technical novelty, compared to other work.
2. Description of important experiment setup is missing.

Overall, this work made an important contribution despite limited technical novelty. I would be worthwhile to include in COLM.